# Wavelet Latent Diffusion (WaLa): Billion-Parameter 3D Generative Model with Compact Wavelet Encodings

## Abstract

Large-scale 3D generative models require substantial computational resources yet often fall short in capturing fine details and complex geometries at high resolutions. We attribute this limitation to the inefficiency of current representations, which lack the compactness required for generative networks to model effectively. To address this, we introduce Wavelet Latent Diffusion (WaLa), a novel approach that encodes 3D shapes into a wavelet-based, compact latent encodings. Specifically, we compress a $256^3$ signed distance field into a $12^3 \times 4$ latent grid, achieving an impressive 2,427× compression ratio with minimal loss of detail. This high level of compression allows our method to efficiently train large-scale generative networks without increasing inference time. Our models, both conditional and unconditional, contain approximately one billion parameters and successfully generate high-quality 3D shapes at $256^3$ resolution. Moreover, WaLa offers rapid inference, producing shapes within 2–4 seconds depending on the condition, despite the model's scale. We demonstrate state-of-the-art performance across multiple datasets, with significant improvements in generation quality, diversity, and computational efficiency. Upon acceptance, we will open-source the code and model weights for public use and reproducibility.

## 1 Introduction

Training generative models on large-scale 3D data presents significant challenges. The cubic nature of 3D data drastically increases the number of input variables the model must manage, far exceeding the complexity found in image and natural language tasks. This complexity is further compounded by storage and streaming issues. Training such large models often requires cloud services, which makes the process expensive for high-resolution 3D datasets, as they take up considerable space and are slow to stream during training. Additionally, unlike other data types, 3D shapes can be represented in various ways, such as voxels, point clouds, meshes, and implicit functions. Each representation presents different trade-offs between quality and compactness. Determining which representation best balances high fidelity with compactness for efficient training and generation remains an open challenge. Finally, 3D representations often exhibit complex hierarchical structures with details at multiple scales, making it challenging for a generative model to capture both global structure and fine-grained details simultaneously.

To address these challenges, current state-of-the-art methods for large generative models typically employ three main strategies. The first involves using low-resolution representations, such as sparse point clouds (Nichol et al., 2022c; Jun & Nichol, 2023b), low-polygon meshes (Chen et al., 2024b), or coarse grids. While these approaches reduce computational complexity, they are limited in their ability to model the full distribution of 3D shapes, struggle to capture intricate details, and often lead to lossy representations. The second approach represents 3D shapes through a collection of 2D images (Yan et al., 2024a) or incorporates images (Hong et al., 2023; Li et al., 2023a; Liu et al., 2024; Xu et al., 2023b) into the training loss. However, this method suffers from long training times due to the need for rendering and fails to capture internal details of 3D shapes, as it primarily focuses on external appearances. The third strategy introduces more compact input representations (Hui et al., 2024; Zhou et al., 2024; Ren et al., 2024; Yariv et al., 2024) to reduce the number of variables the generative model must handle. While these representations simplify the input space,

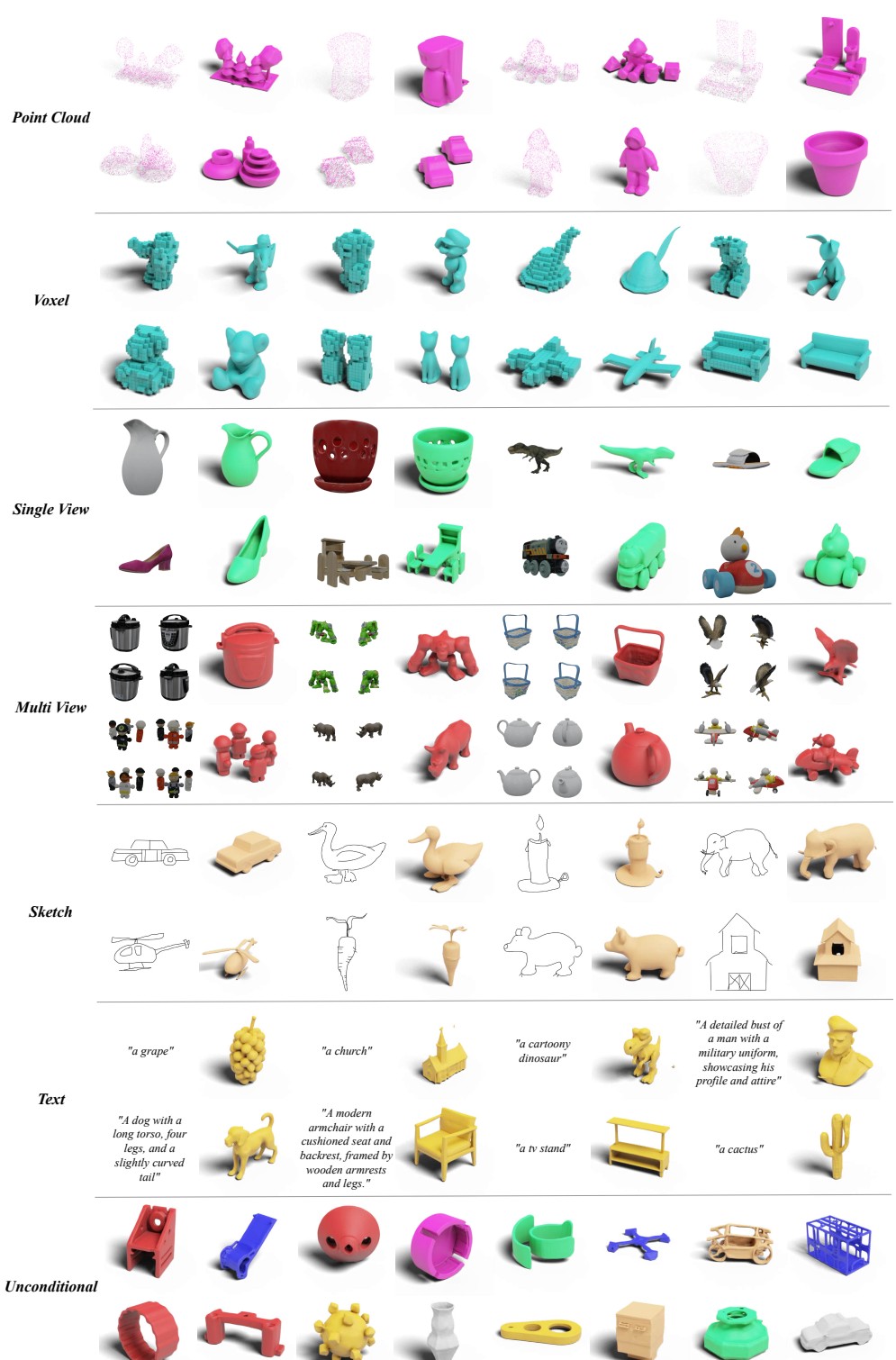

Figure 1: Generation results using WaLa. Compressing 3D shapes into compact latent representations, our method enables efficient training and rapid inference of high-quality 3D shapes at $256^3$ resolution, achieving state-of-the-art performance in both conditional and unconditional settings. Remarkably, WaLa can generate diverse shapes from a variety of conditioning inputs. In the conditional results, even columns show inputs; odd columns show generated shapes(Indexing from 0).

they are often irregular or discrete in nature making it challenging to model using neural networks and can still be relatively large compared to image or natural language data, making it difficult to scale model parameters efficiently.

One prominent compact input representation is wavelet-based representations, which include Neural Wavelet (Hui et al., 2022), UDiFF (Zhou et al., 2024), and wavelet-tree frameworks (Hui et al., 2024). These methods utilize wavelet transforms and their inverses to seamlessly convert between wavelet spaces and high-resolution truncated signed distance function (TSDF) representations. They offer several key advantages: data can be easily compressed by discarding select coefficients with minimal loss of detail, and the interrelationships between coefficients facilitate efficient storage, streaming, and processing of large-scale 3D datasets compared to directly using TSDFs (Hui et al., 2024). However, despite these benefits, wavelet-based representations remain substantially large, especially when scaling up for large-scale generative models. For example, a $256^3$ TSDF can be represented as a wavelet-tree of size $46^3 \times 64$ (Hui et al., 2024), which is equivalent to a $1,440 \times 1,440$ RGB image. Scaling within this space continues to pose significant challenges.

In this work, we further build on the wavelet representation described above. To efficiently scale a generative model, we propose the Wavelet Latent Diffusion (WaLa) framework, where we train an autoencoder to further compress this representation, leading to minimal loss of information. We start by compressing 3D wavelet representations (Hui et al., 2024) using a convolution-based VQ-VAE, reducing a $256^3$ truncated signed distance function (TSDF) to a $12^3 \times 4$ grid. This achieves a 2,427× compression while maintaining an impressive reconstruction IOU (Intersection over Union) of 97.8 on the GSO dataset. As a result, the generative model does not need to model local details and can focus on the global structure. This further enables the training of large-scale 3D generative models with up to a billion parameters, producing highly detailed and diverse shapes. WaLaallows for controlled generation through multiple input modalities without adding many inductive biases, making the framework flexible and not limited to single-view to 3D reconstruction tasks. Consequently, our model generates highly detailed 3D shapes with complex geometry, plausible structures, intricate topologies, and smooth surfaces.

In summary, we make the following contributions:

- We introduce WaLa, a method that tackles the dimensional and computational challenges of 3D generation with impressive compression while maximizing fidelity.

- Our large billion-parameter model generates high-quality 3D shapes within 2-4 seconds, significantly outperforming state-of-the-art benchmarks in 3D shape generation.

- Our model demonstrates exceptional versatility, accepting diverse input modalities such as single/multi-view images, voxels, point clouds, depth data, sketches, and textual descriptions (see Figure 1), making it applicable to a wide range of 3D modeling tasks.

- To facilitate reproducibility and encourage further research in this domain, we commit to releasing our large-scale model, comprising approximately one billion parameters, upon acceptance of this paper.

## 2    RELATED WORK

**Neural Shape Representations.** Deep learning for 3D representations has explored several different representations. Initially, volumetric methods using 3D convolutional networks were employed (Wu et al., 2015; Maturana & Scherer, 2015), but they were limited by resolution and efficiency. The field then advanced to multi-view CNNs that apply 2D processing to rendered views (Su et al., 2015; Qi et al., 2016), and further explored sparse point cloud representations with networks like PointNet and its successors (Qi et al., 2017a;b; Wang et al., 2019). Additionally, neural implicit representations for compact, continuous modeling were developed (Park et al., 2019; Mescheder et al., 2019; Chen & Zhang, 2019). Explicit mesh-based and boundary representations (BREP) have gained attention, enhancing both discriminative and generative capabilities in CAD-related applications (Hanocka et al., 2019; Chen et al., 2024b; Jayaraman et al., 2021; Lambourne et al., 2021). Recently, wavelet representations (Hui et al., 2022; Zhou et al., 2024; Hui et al., 2024) have become very popular. Wavelet decompositions of SDF signals enabled tractable modeling of high-resolution shapes. We extend previous research by addressing the dimensional and computational hurdles of 3D generation. Our novel techniques for efficient shape processing enable high-quality 3D generation at scale, accommodating datasets with millions of shapes.

**3D Generative Models.** 3D generative models have evolved rapidly, initially dominated by Generative Adversarial Networks (GANs)(Goodfellow et al., 2014; Wu et al., 2016). Subsequent advancements integrated differentiable rendering with GANs, utilizing multi-view losses for enhanced fidelity. Parallel developments explored normalizing flows (Yang et al., 2019; Klokov et al., 2020; Sanghi et al., 2022) and Variational Autoencoders (VAEs) (Mo et al., 2019). Additionally, autoregressive models also gained traction for their sequential generation capabilities (Cheng et al., 2022; Nash et al., 2020; Sun et al., 2020; Mittal et al., 2022; Yan et al., 2022; Zhang et al., 2022; Sanghi et al., 2023a). The recent success of diffusion models in image generation has sparked intense interest in their application to 3D contexts. Most current approaches employ a two-stage process: first training a Vector-Quantized VAE (VQ-VAE) on 3D representations such as triplanes (Shue et al., 2023b; Chou et al., 2023; Peng et al., 2020; Reddy et al., 2024; Siddiqui et al., 2024; Chen et al., 2022; Gao et al., 2022b; Shue et al., 2023a), implicit forms (Zhang et al., 2023a; Li et al., 2023b; Cheng et al., 2023), or point clouds (Jun & Nichol, 2023a; Zeng et al., 2022), then applying diffusion models to the resulting latent space. Incorporating autoencoders to process latent spaces allowed for the generation of complex representations like point clouds (Jun & Nichol, 2023a; Zeng et al., 2022) and implicit forms (Zhang et al., 2023a; Li et al., 2023b; Cheng et al., 2023; Zhang et al., 2024). Direct diffusion training on 3D representations, though less explored, has shown promise in point clouds (Nichol et al., 2022a; Zhou et al., 2021; Luo & Hu, 2021; Nakayama et al., 2023), voxels (Zheng et al., 2023), occupancy (Ren et al., 2024), and neural wavelet coefficients (Hui et al., 2022; Liu et al., 2023d; Hui et al., 2024). Our work advances this frontier by bridging the gap between compact representation and high-fidelity generation.

**Conditional 3D Models.** Two primary paradigms dominate conditional 3D generative models, each with its own approach to 3D content creation. The first paradigm ingeniously repurposes large-scale 2D conditional image generators, such as (Rombach et al., 2022a) or Imagen (Saharia et al., 2022), for 3D synthesis. This approach employs a differentiable renderer to project 3D shapes into 2D images, enabling comparison with target images or alignment with text-to-image model distributions(Jain et al., 2022; Michel et al., 2022; Poole et al., 2022). Initially focused on text-to-3D generation, this method has expanded to accommodate various input modalities, including single and multi-view images (Deng et al., 2023; Melas-Kyriazi et al., 2023; Xu et al., 2022; Liu et al., 2023c; Deitke et al., 2023; Qian et al., 2023; Shi et al., 2023; Wang et al., 2023; Liu et al., 2023b), and even sketches (Mikaeili et al., 2023). This approach, while novel, is limited by its computational demands. An alternative paradigm uses dedicated conditional 3D generative models trained on either paired datasets or through zero-shot learning. These paired models show adaptability to various input conditions, ranging from point clouds (Zhang et al., 2022; 2023b) and images (Zhang et al., 2022; Nichol et al., 2022a; Jun & Nichol, 2023a; Zhang et al., 2023b; Chen et al., 2024a; Tang et al., 2024; Li et al., 2023a; Xu et al., 2024) to low-resolution voxels (Chen et al., 2021; 2023b), sketches (Lun et al., 2017; Guillard et al., 2021; Gao et al., 2022a; Kong et al., 2022), and textual descriptions (Nichol et al., 2022a; Jun & Nichol, 2023a). Concurrently, zero-shot methods have gained traction, particularly in text-to-3D (Sanghi et al., 2022; 2023a; Liu et al., 2022; Xu et al., 2023a; Yan et al., 2024b) and sketch-to-3D applications (Sanghi et al., 2023b), showcasing the potential for more flexible and generalizable 3D generation. We expand on the second paradigm, developing a large-scale paired conditional generative model for 3D shapes. This approach enables fast generation without per-instance optimization, supports diverse inputs, and facilitates unconditional generation and zero-shot tasks like shape completion.

## 3 METHOD

Training generative models on large-scale 3D data is challenging because of the data's complexity and size. This has driven the creation of compact representations like neural wavelets, facilitating efficient neural network training. To represent a 3D shape with wavelets, it is first converted into a Truncated Signed Distance Function (TSDF) grid. A wavelet transform is then applied to decompose the TSDF into coarse coefficients ($C_0$) and detail coefficients at various levels ($D_0$, $D_1$, $D_2$). Various wavelet transforms, such as Haar, biorthogonal, or Meyer wavelets, can be employed. Most current methods utilize the biorthogonal wavelet transform (Hui et al., 2022; Zhou et al., 2024; Hui et al., 2024). The coarse coefficients primarily capture the essential shape information, while the detail coefficients represent high-frequency details. To compress this representation, different filtering schemes can be applied to remove certain coefficients, though this involves a trade-off in reconstruction quality. In the neural wavelet representation, all detail coefficients are discarded during the training of the generative model, and a regression network is used to predict the missing de-

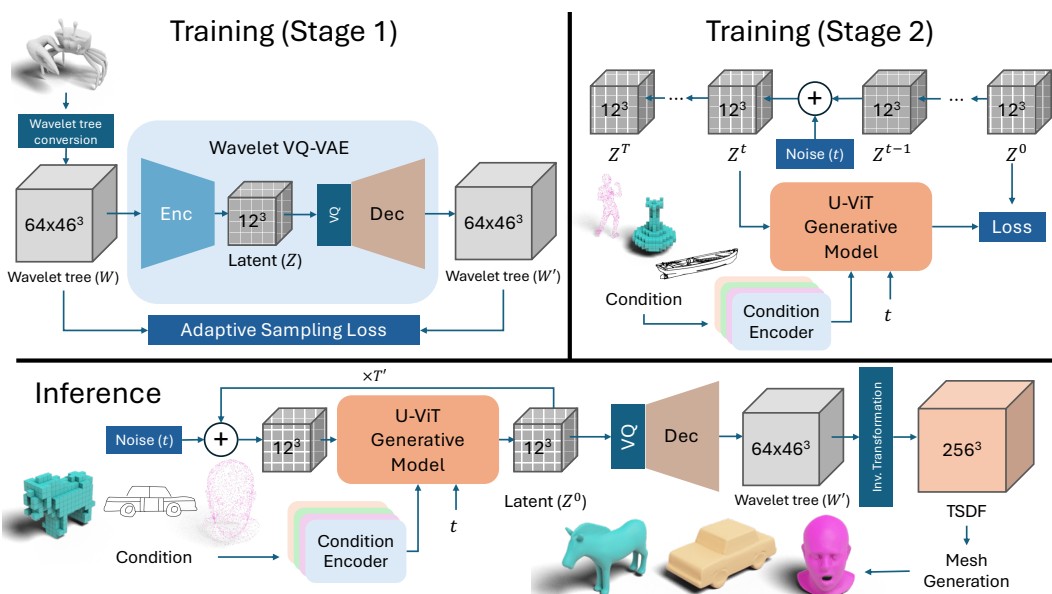

Figure 2: Overview of the WALA network architecture and 2-stage training process and inference method. Top Left: Stage 1 autoencoder training, compressing Wavelet Tree (W) shape representation into a compact latent space. Top Right: Conditional/unconditional diffusion training. Bottom: Inference pipeline, illustrating sampling from the trained diffusion model and decoding the sampled latent into a Wavelet Tree (W), then into a mesh.

tail coefficients $D_0$. In contrast, the wavelet-tree representation retains all coarse coefficients ($C_0$), discards the third level of detail coefficients ($D_2$), and selectively keeps the most significant coefficients from $D_0$ along with their corresponding details in $D_1$, using a subband coefficient filtering scheme. The neural wavelet representation, while modeling a smaller number of input variables, has lower reconstruction quality than the wavelet-tree representation, making the latter a more attractive option.

Building upon these efficient wavelet representations, our method requires a large collection of 3D shapes, denoted as $S = \{(W_n, \Theta_n)\}_{n=1}^N$, where each shape $n$ consists of a diffusible wavelet tree representation $W_n$ (Hui et al., 2024) and an optional associated condition $\Theta_n$. The representation $W_n \in \mathbb{R}^{64 \times 46^3}$ is obtained by converting a TSDF of resolution $256^3$. Depending on the conditional generative model, the condition $\Theta_n$ can be a single-view image, multi-view images, a voxel representation, a point cloud, or multi-view depth maps. The condition $\Theta_n$ may be omitted if the model is unconditional or when training the vector-quantized autoencoder (VQ-VAE). Training our model comprises two stages: in the first step, we train a convolution-based VQ-VAE to encode the diffusible wavelet tree representation into a more compact grid latent space $Z$ using the adaptive sampling loss. After training the VQ-VAE, we obtain a shape latent grid $Z_n$ for each shape in $S$. In the second phase, we train a diffusion-based generative model on this latent grid $Z_n$, which is conditioned on a sequence of condition vectors derived from one of the aforementioned conditions. During inference, we initiate with a completely noisy latent vector and employ the conditional generative network to denoise it progressively through the diffusion process, utilizing classifier-free guidance. The whole process is shown in Figure 2.

### 3.1 STAGE 1: WAVELET VQ-VAE

Our primary objective is to compress the diffusible wavelet tree representation (Hui et al., 2024) into a compact latent space without significant loss of fidelity, thereby facilitating the training of a generative model directly on this latent space. Decoupling compression from generation allows for efficient scaling of a large generative model within the latent space. To this end, we employ a convolution-based VQ-VAE, known for producing sharper reconstructions and mitigating issues

like posterior collapse (Van Den Oord et al., 2017; Razavi et al., 2019; Baykal et al., 2024). Specifically, the encoder $Enc(\cdot)$ maps the input $W_n$ to a latent representation $Z_n = Enc(W_n)$, which is then quantized via a vector quantization layer and decoded by $Dec(\cdot)$ to reconstruct the shape $W_n' = Dec(\text{VQ}(Z_n))$. By integrating the vector quantization layer with the decoder, as in (Rombach et al., 2022b), we ensure that the generative model is trained on pre-quantized latent codes. This approach leverages the robustness of the quantization layer to small perturbations by mapping generated codes to the nearest embeddings in the codebook after generation. Empirical results confirm the effectiveness of this strategy (see Ablation Section C.4).

To train the VQ-VAE, we employ a combination of losses: a reconstruction loss to ensure fidelity between the original and reconstructed shapes, a codebook loss to encourage the codebook embeddings to adapt to the distribution of encoder outputs, and a commitment loss to align the encoder's outputs closely with the codebook embeddings. We apply a reconstruction loss $\mathcal{L}_{\text{rec}}(W_n, W_n')$, during which we adopt a adaptive sampling loss strategy (Hui et al., 2024) to focus more effectively on high-magnitude detail coefficients (i.e., $D_0$ and $D_1$) while still considering the others. Since most detail coefficients are low in magnitude and contribute minimally to the overall shape quality, this approach identifies significant coefficients in each subband based on their magnitude relative to the largest coefficient, forming a set $P_0$ of important coordinates. By structuring the training loss to emphasize these crucial coefficients and incorporating random sampling of less important ones, the model efficiently concentrates on key information without neglecting finer details. This is formalized in the equation below:

$$\mathcal{L}_{\text{rec}} = L_{\text{MSE}}(C_0, C_0') + \frac{1}{2} \sum_{D \in \{D_0, D_1\}} [L_{\text{MSE}}(D[P_0], D'[P_0]) + L_{\text{MSE}}(R(D[P_0']), R(D'[P_0']))] \quad (1)$$

In this context, $L_{\text{MSE}}(X, Y)$ denotes the mean squared error between $X$ and $Y$. The coefficients $C_0, D_0, D_1$ extracted from $W_n$ represent the coarse and detail components, respectively, while their reconstructed counterparts $C_0', D_0', D_1'$ are derived from $W_n'$. The notation $D[P_0]$ refers to the coefficients in $D$ at the positions specified by the set $P_0$, with $P_0'$ being its complement. The function $R(D[P_0'])$ randomly selects coefficients from $D[P_0']$ such that the number of selected coefficients equals $|P_0|$. By balancing the number of coefficients in the last two terms of the loss function, we emphasize critical information while regularizing less significant coefficients through random sampling. This approach is also empirically validated in Ablation (Section C.1).

Our model is trained on 10 million samples from 19 datasets; however, a substantial portion of this data is skewed toward simple CAD objects. To address this imbalance, once the VQ-VAE model has converged, we further fine-tune it using a simple strategy that employs equal amounts of data from each of the 19 datasets. Empirically, we find that this approach enhances reconstruction results, as demonstrated in Ablation (Section C.2).

## 3.2 STAGE 2: LATENT DIFFUSION MODEL

In the second stage, we train a large-scale generative model with billions of parameters on the latent grid, either as an unconditioned model to capture the data distribution or conditioned on diverse modalities $\Theta_n$ (e.g., point clouds, voxels, images). We use a diffusion model within the Denoising Diffusion Probabilistic Models (DDPM) framework (Ho et al., 2020), modeling the generative process as a Markov chain with two phases.

First, the forward diffusion process gradually adds Gaussian noise to the initial latent code $Z_n^0$ over $T$ steps, resulting in $Z_n^T \sim \mathcal{N}(0, I)$. Then, the reverse denoising process employs a generator network $\theta$, conditioned on $\Theta_n$, to systematically remove the noise and reconstruct $Z_n^0$. The generator predicts the original latent code $Z_n^0$ from any intermediate noisy latent code $Z_n^t$ at time step $t$, using $f_\theta(Z_n^t, t, \Theta_n) \approx Z_n^0$, and is optimized using a mean-squared error loss:

$$\mathcal{L} = \mathbb{E}\left[\|f_\theta(Z_n^t, t, \Theta_n) - Z_n^0\|^2\right]$$

Here, $Z_n^t$ is obtained by adding Gaussian noise $\epsilon$ to $Z_n^0$ at step $t$ using a cosine noise schedule (Dhariwal & Nichol, 2021). The condition $\Theta_n$ is a latent set of vectors derived from various conditioning modalities. It is injected into the U-ViT generator (Hoogeboom et al., 2023) using cross-attention

and by modulating the normalization parameters in the ResNet and cross-attention layers, as described in (Esser et al.). This is achieved via a conditional encoder for different modalities. During training, we apply a small dropout to the condition to implement classifier-free guidance during inference. In the case of unconditional generation, no conditioning is applied. For most input conditions (point clouds, voxels, images, multi-view images, multi-view depth), we directly train a conditional generative model. For the sketch condition, we take the image-conditioned generative model and fine-tune it with synthetic sketch data. For text-to-3D, we fine-tune an MVDream (Xu et al., 2023b) to generate multi-view depth, as it provides better reconstruction than multi-view images (see experiment Section 4.2.3), and then use our model during inference. Further details are provided in the appendix.

## 3.3 INFERENCE

At test time, we begin with a fully noisy latent vector $Z_n^T \sim \mathcal{N}(0, I)$ and iteratively denoise it to reconstruct the original latent code $Z_n^0$ through the reverse diffusion process, as described in DDPM. For conditional generation, we apply classifier-free guidance (Ho & Salimans, 2022) by interpolating between the unconditional and conditional denoising predictions, steering the generation process toward the desired output. This approach allows for greater control over the quality-diversity trade-off. Once the final latent code $Z_n^0$ is obtained, we use the pre-trained decoder network from the first stage to generate the final 3D shape in wavelet form. Subsequently, we apply the inverse wavelet transform to obtain the final 3D shape as an SDF. The SDF can be converted to a mesh using marching cubes. Notably, we can generate multiple samples for the same conditional input by using different initializations of the noisy latent vector.

## 4 RESULTS

### 4.1 EXPERIMENTAL SETUP

**Datasets.** Our training data is a massive dataset of over 10 million 3D shapes, assembled from 19 publicly available sub-datasets, including ModelNet Vishwanath et al. (2009), ShapeNet Chang et al. (2015), SMLP Loper et al. (2015), Thingi10K Zhou & Jacobson (2016), SMAL Zuffi et al. (2017), COMA Ranjan et al. (2018), House3D Wu et al. (2018), ABC Koch et al. (2019), Fusion 360 Willis et al. (2021), 3D-FUTURE Fu et al. (2021), BuildingNet Selvaraju et al. (2021), DeformingThings4D Li et al. (2021), FG3D Liu et al. (2021), Toys4K Stojanov et al. (2021), ABO Collins et al. (2022), Infinigen Raistrick et al. (2023), Objaverse Deitke et al. (2023), and two subsets of ObjaverseXL Deitke et al. (2023) (Thingiverse and GitHub). These sub-datasets target specific object categories: for instance, CAD models (ABC and Fusion 360), furniture (ShapeNet, 3D-FUTURE, ModelNet, FG3D, ABO), human figures (SMLP and DeformingThings4D), animals (SMAL and Infinigen), plants (Infinigen), faces (COMA), and houses (BuildingNet, House3D). Additionally, Objaverse and ObjaverseXL provide a wider variety of generic objects sourced from the internet, covering the aforementioned categories and other diverse objects. As mentioned in Hui et al. (2024), each sub-dataset was split into two portions for data preparation: 98% of the shapes were allocated for training, and the remaining 2% for testing. The final training and testing sets were created by merging the corresponding portions from each sub-dataset. Note that we use the entire testing dataset solely for autoencoder reconstruction validation. We also apply 90-degree rotation augmentation along each axis, doing the same for the corresponding conditions (point clouds, voxels). We also create a balanced training set across these 19 datasets by sampling 10,000 shapes from each. If a dataset contains fewer than 10,000 shapes, we duplicate the data until the target size is reached.

**Training Details.** We train our VQ-VAE and generative model using the Adam optimizer Kingma & Ba (2014) with a learning rate of 0.0001 and a gradient clipping value of 1. For VQ-VAE training, we use a batch size of 256 with 1,024 codebook embeddings of dimension 4. We train the network until it converges and then fine-tune this autoencoder using a more balanced dataset until it also converges. For the generative model, we use a batch size of 64 and train it for 2–4 million iterations for each modality. Generative models are trained on a single H100 GPU for each condition. We train our model on seven conditions: point cloud with 2,500 points, voxel at $16^3$, single-view, multi-view, unconditional, multi-view depth with 4 views, and multi-view depth with 6 views.

**Evaluations Dataset.** We perform qualitative and quantitative evaluation of our method on Google Scanned Objects (GSO) (Downs et al., 2022) and MAS validation data (Hui et al., 2024). Impor-

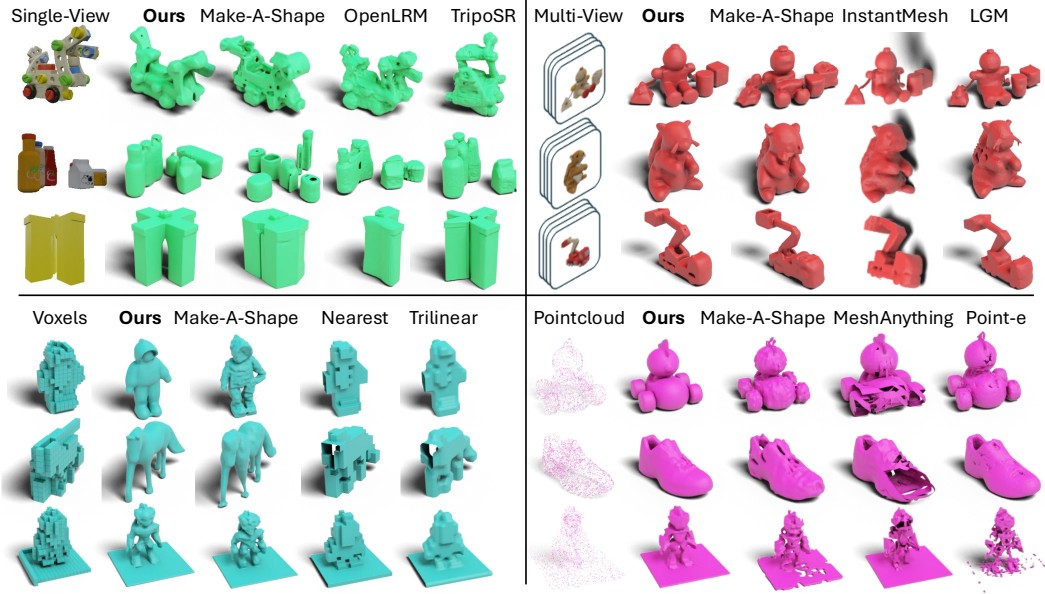

Figure 3: Qualitative comparison with other methods for single-view (top-left), multi-view (top-right), voxels (bottom-left), and point cloud (bottom-right) conditional input modalities.

tantly Google Scanned Objects (GSO) is not part of the massive dataset mentioned above(Ref 4.1) used to train our model. Consequently, evaluating on Google Scanned Objects (GSO) data assesses the cross-domain generalization of our method. We include all validation objects from the GSO dataset to ensure a broad evaluation. MAS validation data is the unseen test set consisting of 50 randomly selected shapes from the large-scale compiled dataset. This ensures that validation data contains all the subcategories like CAD models, human figures, faces, houses, and others, thereby enabling a comprehensive evaluation. We present three metrics for each method on both datasets, the metrics being: (i)Light Field Distance (LFD)(Chen et al., 2003) which evaluates how alike two 3D models appear when viewed from multiple angles. (ii) Intersection over Union (IoU) ratio, which compares the intersection volume to the total volume of two voxelized 3D objects. and (iii) Chamfer Distance (CD), which measures the similarity between two shapes based on the minimum distance between corresponding points on their surfaces.

## 4.2 EVALUATION

We conducted a comprehensive study across various modalities, quantitatively evaluating our method against baselines using four distinct input types: point clouds, voxels, single-view images, and multi-view images. For qualitative analysis, we present the results of all our models, showcasing select visual outcomes in Figure 1 and providing additional examples in the appendix. We also include a detailed ablation study in the appendix.

### 4.2.1 POINT CLOUD-TO-3D

In this experiment, we aim to generate a SDF from an input point cloud. We show qualitative results on this task in Fig. 3. To quantitatively assess WaLa's performance, we compare it against both traditional and large scale data-centric techniques in Tab. 1. First, We benchmark against Poisson surface reconstruction a traditional approach that uses a heuristic method to create smooth meshes from point clouds. We estimate normals using 5 nearest neighbour points using O3D (Zhou et al., 2018). Following Poisson surface reconstruction, we eliminate vertices whose density values fall below the 0.2 quantile to avoid spurious faces. Additionally, we evaluate our method alongside data-driven generative model like Point-E Nichol et al. (2022b). We use a Point-E version which contains a SDF network fine-tuned to estimated the distance field. We also compare our method with MeshAnything, a transformer-based neural network designed for meshing. For a fair evalua-

Table 1: Quantitative comparison between different methods of point cloud to mesh generation. We present LFD, IOU and CD metrics. Our method outperforms the other methods on both GSO and MAS Validation datasets.

| Method | GSO Dataset | | | MAS Dataset | | |
|---|---|---|---|---|---|---|
| | LFD ↓ | IoU ↑ | CD ↓ | LFD ↓ | IoU ↑ | CD ↓ |
| Poisson surface reconstruction (Kazhdan et al., 2006) | 3306.66 | 0.3838 | 0.0055 | 4565.56 | 0.2258 | 0.0085 |
| Point-E SDF model (Nichol et al., 2022c) | 2301.96 | 0.6006 | 0.0037 | 4378.51 | 0.4899 | 0.0158 |
| MeshAnything (Chen et al., 2024b) (2500 points) | 2228.62 | 0.3731 | 0.0064 | 2892.13 | 0.3378 | 0.0091 |
| MeshAnything (Chen et al., 2024b) (8192 points) | 2393.43 | 0.4316 | 0.0096 | 2931.36 | 0.3857 | 0.0102 |
| Make-A-Shape (Hui et al., 2024) | 2274.92 | 0.7769 | 0.0019 | 1857.84 | 0.7595 | 0.0036 |
| WaLa(Ours) | **1114.01** | **0.9389** | **0.0011** | **1467.55** | **0.8625** | **0.0014** |

Table 2: Quantitative evaluation on lower resolution voxel data to mesh generation task. Our method's performance surpasses traditional Nearest neighbour and Trilinear upsampling as well as data-centric method like Make-a-Shape.

| Method | GSO Dataset | | | MAS Dataset | | |
|---|---|---|---|---|---|---|
| | LFD ↓ | IoU ↑ | CD ↓ | LFD ↓ | IoU ↑ | CD ↓ |
| Nearest Neighbour Interpolation | 5158.63 | 0.1773 | 0.0225 | 5401.12 | 0.1724 | 0.0217 |
| Trilinear Interpolation | 4666.85 | 0.1902 | 0.0361 | 4599.97 | 0.1935 | 0.0371 |
| Make-A-Shape (Hui et al., 2024) | 1913.69 | 0.7682 | 0.0029 | 2566.22 | 0.6631 | 0.0051 |
| WaLa(Ours) | **1544.67** | **0.8285** | **0.0020** | **1874.41** | **0.75739** | **0.0020** |

tion, we follow their procedure by using ground-truth normals instead of estimating normals from point cloud data. All methods are evaluated using 2,500 uniformly sampled points, and we further present MeshAnything's performance with 8,192 points for comparison. In terms of IoU on the GSO dataset, Point-E, MeshAnything (8192), and Make-A-Shape achieve scores of 0.60, 0.43, and 0.77, respectively. On the MAS validation dataset, they reach 0.48, 0.38, and 0.75. We attribute MeshAnything's underperformance compared to Point-E and Make-A-Shape to the scale of the data it was trained on. Our model significantly outperforms these baselines, achieving IoU scores of 0.93 on the GSO dataset and 0.86 on the MAS validation dataset, representing a notable relative improvement of 21% and 15%. Similarly, our method surpasses the baselines on LFD and CD metrics as well. These results demonstrate that our approach consistently excels in the point cloud-to-3D task across various object types.

### 4.2.2 VOXEL-TO-3D

We investigate using low-resolution voxels as input to our model to generate a Signed Distance Function (SDF) that reconstructs the object's geometry. In Tab.2 and Fig. 3, we present the result on low-resolution Voxel-to-3D task. We evaluate our method against conventional techniques for converting low-resolution voxels into meshes. For the baseline comparisons, we apply interpolation methods like nearest neighbor and trilinear interpolation, then use the marching cubes (Lorensen & Cline, 1998) algorithm to generate the meshes. The qualitative results for both resolutions are shown in the figure. From this analysis, we observe that our method consistently generates smooth and clean surfaces. Even in cases with ambiguity, particularly at the $16^3$ voxel resolution, our approach produces plausible shapes maintaining strong performance across different complexities. We present quantitative results in Tab. 2, we discuss these results further in the appendix.

### 4.2.3 IMAGE-TO-3D

Our experiment compares WaLa with other state-of-the-art image-to-3D generative models, focusing on both single-view and multi-view scenarios. In the single-view setting, our model generates 3D shapes from a single input image. For multi-view generation, we leverage four images along with their corresponding camera parameters. This dual approach allows us to evaluate the model's performance across varying conditions, demonstrating the versatility and effectiveness of our generative model in different image-to-3D generation contexts. Our results on the GSO and MAS validation datasets are shown in Tab. 3 and Fig. 3. In Tab. 3 we present quantitative results for the Image-to-3D task at the top and the Multiview-to-3D task at the bottom. As demonstrated, our method consistently outperforms the other 3D generation techniques across both tasks.

Table 3: Comparison between different methods on Image-to-3D task (Top) and Multiview-to-3D task (Bottom). Quantitative evaluation shows that our single-view model excels the baselines, achieving the highest IoU and lowest LFD metrics. Our multi-view model further enhances performance by incorporating additional information. RGB 4, Depth 4, and Depth 6 represents conditioning using RGB images from 4 different views, and depth estimates from 4 and 6 views respectively.

| | Method | Inference Time↓ | GSO Dataset | | | MAS Val Dataset | | |
|---|---|---|---|---|---|---|---|---|
| | | | LFD ↓ | IoU ↑ | CD ↓ | LFD ↓ | IoU ↑ | CD ↓ |
| Single-view | Point-E (Nichol et al., 2022a) | ∼31 Sec | 5018.73 | 0.1948 | 0.02231 | 6181.97 | 0.2154 | 0.03536 |
| | Shap-E (Jun & Nichol, 2023a) | ∼6 Sec | 3824.48 | 0.3488 | 0.01905 | 4858.92 | 0.2656 | 0.02480 |
| | One-2-3-45 (Liu et al., 2023a) | ∼45 Sec | 4397.18 | 0.4159 | 0.04422 | 5094.11 | 0.2900 | 0.04036 |
| | OpenLRM (He & Wang, 2024) | ∼5 Sec | 3198.28 | 0.5748 | **0.01303** | 4348.20 | 0.4091 | 0.01668 |
| | TripoSR(Tochilkin et al., 2024) | ∼1 Sec | 3750.65 | 0.4524 | 0.01388 | 4551.29 | 0.3521 | 0.03339 |
| | InstantMesh(Xu et al., 2024) | ∼10 Sec | 3833.20 | 0.4587 | 0.03275 | 5339.98 | 0.2809 | 0.05730 |
| | LGM(Tang et al., 2024) | ∼37 Sec | 4391.68 | 0.3488 | 0.05483 | 5701.92 | 0.2368 | 0.07276 |
| | Make-A-Shape(Hui et al., 2024) | ∼2 Sec | 3406.61 | 0.5004 | 0.01748 | 4071.33 | 0.4285 | 0.01851 |
| | WaLa(RGB) | ∼2.5 Sec | **2509.20** | **0.6154** | 0.02150 | **2920.74** | **0.6056** | **0.01530** |
| Multi-view | InstantMesh(Xu et al., 2024) | ∼1.5 Sec | 3009.19 | 0.5579 | 0.01560 | 4001.09 | 0.4074 | 0.02855 |
| | LGM(Tang et al., 2024) | ∼35 Sec | 1772.98 | 0.6842 | 0.00783 | 2712.30 | 0.5418 | 0.00867 |
| | Make-A-Shape(Hui et al., 2024) | ∼2 Sec | 1890.85 | 0.7460 | 0.00337 | 2217.25 | 0.6707 | 0.00350 |
| | WaLa(RGB 4) | ∼2.5 Sec | 1260.64 | 0.8500 | 0.00182 | 1540.22 | 0.8175 | 0.00208 |
| | WaLa(Depth 4) | ∼4 Sec | 1185.39 | 0.87884 | 0.00164 | 1417.40 | 0.83313 | 0.00160 |
| | WaLa(Depth 6) | ∼4 Sec | **1122.61** | **0.91245** | **0.00125** | **1358.82** | **0.85986** | **0.00129** |

For the Image-to-3D task, Point-E, considered a baseline 3D generation method, achieves IoU scores of 0.19 on the GSO dataset and 0.24 on the MAS validation data. Other methods improve over it, with recent methods like OpenLRM, TripoSR, InstantMesh, LGM and Make-A-Shape reaching IoU scores of (0.57, 0.40), (0.45, 0.35), (0.45, 0.28), (0.34, 0.23) and (0.50, 0.42). WaLa sets a new state-of-the-art, achieving IoU scores of 0.61 and 0.60 on the GSO and MAS validation datasets respectively. This represents a 7% improvement over Make-a-shape on GSO and a 41% improvement on MAS. It's important to note that among the metrics, only LFD is rotation-invariant. OpenLRM outputs maintain rotation consistency due to camera parameter considerations, while IoU and CD are sensitive to alignment. While WaLa's CD performance is comparable to OpenLRM, it significantly outperforms OpenLRM on the LFD metric. Similarly, on the Multiview-to-3D task InstantMesh, LGM and Make-A-Shape reach a score of (0.55, 0.40), (0.68, 0.54) and (0.74, 0.67). WaLa, when conditioned on RGB images, outperforms competing methods, achieving IoU scores of 0.85 and 0.81, representing a relative improvement of 7% on the GSO dataset and 17% on the MAS validation data compared to Make-A-Shape. Notably, WaLa conditioned on depth data(depth can be estimate from RGB images using a off-the-shelf method like AdaBins (Bhat et al., 2020)) surpasses the RGB version, achieving IoU scores of 0.91 and 0.85, offering a 7% and 5% improvement over the RGB-based version, and a 22% and 28% improvement compared to Make-A-Shape. Further, WaLa, conditioned on both RGB and depth maps, outperforms InstantMesh, LGM, and Make-A-Shape in the Multiview-to-3D task, both in LFD and CD metrics. This further highlights our method's ability to generate objects across a wide range of categories.

## 5  CONCLUSION

In this work, we introduced Wavelet Latent Diffusion (WaLa), a novel approach to 3D generation that tackles the challenges of high-dimensional data representation and computational efficiency. Our method compresses 3D shapes into a wavelet-based latent space, enabling highly efficient compression while preserving intricate details. WaLa marks a significant leap forward in 3D shape generation, with our billion-parameter model able to generate high-quality shapes in just 2–4 seconds, outperforming current state-of-the-art methods. Its versatility allows it to handle diverse input modalities, including single and multi-view images, voxels, point clouds, depth maps, sketches, and text descriptions, making it adaptable to a wide range of 3D modeling tasks. We believe WaLa sets a new benchmark in 3D generative modeling by combining efficiency, speed, and flexibility. Upon acceptance, we will release our model and code to promote further research and support reproducibility within the community.

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
