# Wavelet Latent Diffusion (WaLa): Billion-Parameter 3D Generative Model with Compact Wavelet Encodings

## A  Additional Results and Details: https://anoniclr2025.github.io

Please refer to `https://anoniclr2025.github.io` for more results and details regarding our model.

## B  Architecture details

In the first stage, we train a convolution-based VQ-VAE using a codebook size of 1024 with a dimension of 4. We downsample the input wavelet tree representation to a $12^3 \times 4$ latent space. Our generative model operates within this latent space by utilizing the U-ViT architecture Hoogeboom et al. (2023), incorporating two notable modifications. Firstly, we do not perform any additional downsampling since our latent space is already quite small. Instead, the model comprises multiple ResNet blocks followed by attention blocks, and then more ResNet blocks at the end, with a skip connection from the initial ResNet block. The attention blocks include both self-attention and cross-attention mechanisms, as described in (Chen et al., 2023). Secondly, we modulate the layer normalization parameters in both the ResNet and attention layers, following the approach detailed in (Esser et al.). This tailored architecture enables our generative model to effectively operate within the compact latent space, enhancing both performance and efficiency.

In this section, we describe the details of the various conditions utilized in our model:

1. **Point Cloud**: During training, we randomly select 2,500 points from the pre-computed pointcloud dataset. These points are encoded into feature vectors using the PointNet encoder Qi et al. (2017). To aggregate these feature vectors into a condition latent vectors, we apply attention pooling as described in Lee et al. (2019). This process converts the individual points into a latent set vector. Finally, we pass this latent set vector through additional Multi-Layer Perceptron (MLP) layers to obtain the final condition latent vectors.

2. **Voxel** $16^3$: For voxel-based conditions, we employ a ResNet-based convolutional encoder to process the $16^3$ voxel grid. After encoding, the voxel volume is downsampled once to reduce its dimensionality, resulting in the condition latent vectors. This approach leverages the spatial hierarchy captured by the ResNet architecture to effectively encode volumetric data.

3. **Single View Image Condition**: Our dataset comprises a predetermined set of views for each object. During training, we randomly select one view from this set. This selected view is then passed through the DINO v2 encoder  (Oquab et al., 2023) to extract feature representations. The output from the encoder serves as the condition latent vectors, encapsulating the visual information from the single view.

4. **Sketch**: We initialize the model using the same architecture described in the single view section. Once the base model has converged, we fine-tune it using sketch data. This fine-tuning process involves training the model on sketch representations to adapt the latent vectors to capture the abstract and simplified features inherent in sketches.

5. **Multi-View Image/Depth Condition**: For multi-view scenarios, we select four views for the multi-view RGB image model and two models of four and six views for the multi-view

Table 1: Ablation study on adaptive sampling as well finetuning of the VQ-VAE model.

| Sampling Loss | Amount of finetune data | IOU ↑ | MSE ↓ | D-IOU ↑ | D-MSE ↓ |
|---|---|---|---|---|---|
| No[1] | - | 0.91597 | 0.00270 | 0.91597 | 0.00270 |
| Yes[1] | - | **0.92619** | **0.00136** | **0.91754** | **0.00229** |
| Yes | - | 0.95479 | 0.00090 | 0.94093 | 0.00169 |
| Yes | 2500 | 0.95966 | 0.00078 | 0.94808 | 0.00149 |
| Yes | 5000 | 0.95873 | 0.00078 | 0.94793 | 0.00149 |
| Yes | 10000 | **0.95979** | **0.00078** | **0.94820** | **0.00148** |
| Yes | 20000 | 0.95707 | 0.00079 | 0.94659 | 0.00150 |

depth model. These views are chosen from pre-selected viewpoints to ensure comprehensive coverage of the entire object. Each selected view is processed individually through the DINO v2 encoder (Oquab et al., 2023), generating a sequence of latent vectors for each view. Finally, the latent vectors from all views are concatenated to form the final conditional latent vectors, effectively integrating information from multiple perspectives.

# C ABLATION STUDIES

## C.1 VQ-VAE ADAPTIVE SAMPLING LOSS ANALYSIS

In this section, we evaluate the importance of adaptive sampling loss by training two autoencoder models for up to 200,000 iterations: one incorporating the adaptive sampling loss and one without it. The results are presented in the first two columns of Table 1 . We use Intersection over Union (IoU) and Mean Squared Error (MSE) to measure the average reconstruction quality across all data points. Additionally, we introduce D-IoU and D-MSE metrics, which assess the average reconstruction performance by weighting each dataset equally. This approach ensures that any data imbalance is appropriately addressed during evaluation.

As shown in the table, even after approximately 200,000 iterations, the model utilizing adaptive sampling loss significantly outperforms the one without it. Specifically, the adaptive sampling loss leads to higher IoU and lower MSE values, indicating more accurate and reliable reconstructions. These results clearly demonstrate the substantial benefits of using adaptive sampling loss in enhancing the performance and robustness of autoencoder models.

## C.2 VQ-VAE ANALYSIS AND FINETUNING ANALYSIS

In this section, we examine the benefits of performing balanced finetuning, as described in the main section of the paper. We conduct an ablation study to determine the optimal amount of finetuning data required per dataset to achieve the best results. The results are presented in the rows following the first two in Table 2 , utilizing the metrics described above.

Our observations indicate that even a small amount of finetuning data improves the IoU and MSE. Specifically, incorporating as few as 2,500 samples per dataset leads to noticeable enhancements in reconstruction accuracy. However, we found that increasing the finetuning data to 10,000 samples per dataset provides optimal performance. At this level, both IOU and Mean Squared Error (MSE) metrics reach their best values, demonstrating the effectiveness of balanced finetuning in enhancing model performance.

Moreover, the D-IoU and D-MSE metrics confirm that using 10,000 samples per dataset effectively mitigates data imbalance to a certain degree. Based on these findings, all subsequent results in this study are based on using 10,000 finetuning samples per dataset. We believe that an interesting area for future work is to improve data curation to further enhance reconstruction accuracy.

## C.3 ARCHITECTURE ANALYSIS OF GENERATIVE MODEL

In this section, we conduct an extensive study on the architectural design choices of the generative model. Given the high computational cost of training large-scale generative models, we implement

---

[1]Results for the first two rows are based on 200k iterations.

Table 2: Ablation study on the generative model design choices.

| Architecture | hidden dim | No. of layers | post or pre | LFD ↓ | IoU ↑ | CD ↓ |
|---|---|---|---|---|---|---|
| U-VIT | 384 | 32 | pre | 1523.74 | 0.8211 | 0.001544 |
| U-VIT | 768 | 32 | pre | 1618.73 | 0.7966 | 0.001540 |
| U-VIT | 1152 | 8 | pre | 1596.88 | 0.8020 | 0.001561 |
| U-VIT | 1152 | 16 | pre | 1521.81 | **0.8237** | 0.001573 |
| U-VIT | 1152 | 32 | pre | **1507.43** | 0.8199 | **0.001482** |
| DiT | 1152 | 32 | pre | 1527.16 | 0.8145 | 0.001602 |
| U-VIT | 1152 | 32 | post | 1576.07 | 0.8176 | 0.001695 |

early stopping after 400,000 iterations. The results are presented in Table 2 . First, we examine the importance of the hidden dimension in the attention layer. It is clearly observed that increasing the dimension enhances performance. A similar trend is noted when additional layers of attention blocks are incorporated. Although the improvement is not pronounced, it is important to mention that these observations are based on only 400,000 iterations. Finally, we compare the DiT (Peebles & Xie, 2023) architecture to the U-ViT architecture (Hoogeboom et al., 2023) and find that U-ViT outperforms DiT. This comparison highlights the superior performance of the U-ViT architecture in our generative model framework.

### C.4 PRE QUANT VS POST QUANT

In this section, we compare whether it is better to apply the generative model to the grid before or after quantization. We conduct this comparison over 400,000 iterations. The results are shown in Table 2. These results indicate that pre-quantization performs better.

## D SKETCH DATA GENERATION

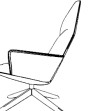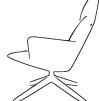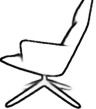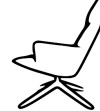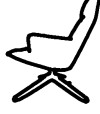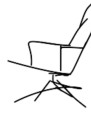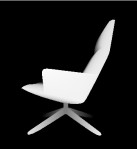

Figure 1: The 6 different sketch types. From left to right: Grease Pencil, Canny, HED, HED+potrace, HED+scribble, CLIPaasso, and a depth map for reference. Mesh taken from (Fu et al., 2021).

We generate sketches using 6 different techniques. In the first technique we use use Blender to perform non-photorealistic rendering of the meshes using a Grease Pencil Line Art modifier. The modifier is configured to use a line thickness of 2 with a crease threshold of 140°. Since disconnected faces can cause spurious lines using this method, we automatically merge vertices by distance using a threshold of 1e-6 before rendering. The second technique takes previously generated depth maps and produces sketches using Canny edge detection. We apply the Canny edge filter built into `imagemagick` using a value of 1 for both the blur radius and sigma and a value of 5% for both the low and high threshold. We then clean the output by running it through the `potrace` program with the flags `--turdsize=10` and `--opttolerance=1`. The third technique uses HED (Xie & Tu, 2015) in its default configuration, also on depth maps. The fourth technique applies `potrace` on top of default HED, and the fifth applies HED's "scribble" filter instead. The sixth and final technique uses CLIPasso (Vinker et al., 2022) on previously rendered color images. We configure CLIPasso to use 16 paths, a width of 0.875, and up to 2,000 iterations, with early stopping if the difference in loss is less than 1e-5.

For the first 5 techniques, sketches are generated from a total of 8 views: the 4 views used for multiview to 3D, plus views from the front, right side, left side, and back. For CLIPasso we only generate sketches from the front, right side, and left back. Aditionally, we only generate sketches from a subset of the 10 million shapes which we constructed by taking up to 10,000 shapes from each of the 20 datasets.

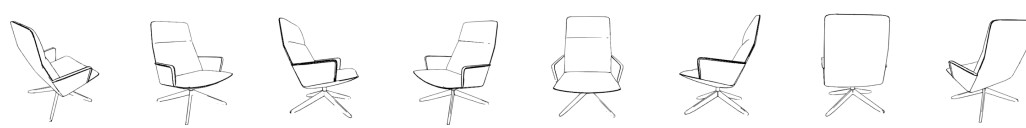

Figure 2: The 8 different views for which sketches were generated. Images created using the Grease Pencil technique on a mesh taken from Fu et al. (2021). The CLIPasso technique was only used on the first, fifth, and sixth views from the left.

During training we augment the sketches by adding random translation, rotation, and scale in order to improve the model's over-sensitivity to line thickness and padding. We also add random positional noise to the shapes in the SVG drawings produced by CLIPasso and `potrace`. Finally, we add a non-affine cage transformation by dividing the image into 9 squares of equal size. We treat the four corners of the central square as control points and move each one independently, warping the image.

## E  TEXT-TO-3D DETAILS

We began by generating captions for our entire dataset using the Internvl 2.0 model (Chen et al., 2024). For each object, we provided four renderings to the model and created two versions of captions by applying two distinct prompts. These initial captions were then augmented using LLaMA 3.1 (Dubey et al., 2024) to enhance their diversity and richness.

Next, we fine-tuned the Stable Diffusion model, initializing it with weights from MVDream (Shi et al., 2023). Utilizing the depth map-text paired data we had collected, we generated six depth maps for each object. To ensure consistency, we identified a uniform cropping box around each object across all depth maps and applied this cropping uniformly to all images. Following the MVDream methodology, we resized the cropped images to 256×256256×256 pixels and employed bfloat16 precision for processing.

During the inference phase, we input text prompts to generate six corresponding depth maps. These depth maps were then used to condition our multi-view depth model, which successfully generated the 3D shape of each object.

## F  MORE VISUAL RESULTS

In Figure 3, we present additional text-to-3D generation results, showcasing the diversity and quality of outputs produced by our model. Each result highlights the model's ability to capture various object details and structures based solely on text prompts. In Figure 4, we illustrate the variety in generation for each caption. For each given caption, we display four different generated outputs, demonstrating the model's capacity to create diverse yet semantically consistent results based on the same input description. These figures collectively emphasize the robustness and versatility of our approach in generating 3D content from textual inputs.

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

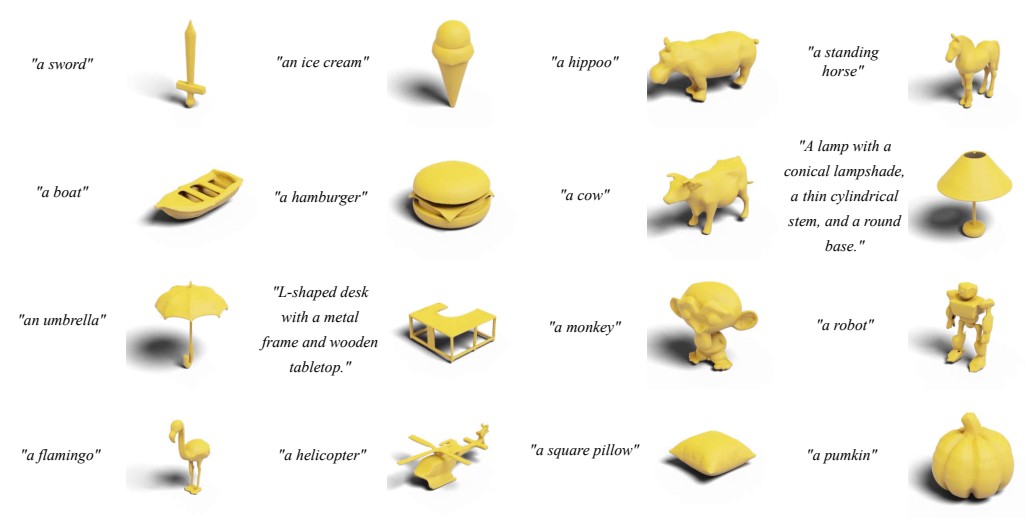

Figure 3: This figure presents more results from the text-to-3D generation task. Each row corresponds to a unique text prompt, with the resulting 3D renderings highlighting the model's capability to produce detailed and varied shapes from these inputs.

Abhimanyu Dubey, Abhinav Jauhri, Abhinav Pandey, Abhishek Kadian, Ahmad Al-Dahle, Aiesha Letman, Akhil Mathur, Alan Schelten, Amy Yang, Angela Fan, et al. The llama 3 herd of models. *arXiv preprint arXiv:2407.21783*, 2024.

Patrick Esser, Sumith Kulal, Andreas Blattmann, Rahim Entezari, Jonas Müller, Harry Saini, Yam Levi, Dominik Lorenz, Axel Sauer, Frederic Boesel, et al. Scaling rectified flow transformers for high-resolution image synthesis, march 2024. *URL http://arxiv. org/abs/2403.03206*.

Huan Fu, Rongfei Jia, Lin Gao, Mingming Gong, Binqiang Zhao, Steve Maybank, and Dacheng Tao. 3d-future: 3d furniture shape with texture. *International Journal of Computer Vision*, 129: 3313–3337, 2021.

Emiel Hoogeboom, Jonathan Heek, and Tim Salimans. simple diffusion: End-to-end diffusion for high resolution images. *arXiv preprint arXiv:2301.11093*, 2023.

Juho Lee, Yoonho Lee, Jungtaek Kim, Adam Kosiorek, Seungjin Choi, and Yee Whye Teh. Set transformer: A framework for attention-based permutation-invariant neural networks. In *International conference on machine learning*, pp. 3744–3753. PMLR, 2019.

Maxime Oquab, Timothée Darcet, Théo Moutakanni, Huy Vo, Marc Szafraniec, Vasil Khalidov, Pierre Fernandez, Daniel Haziza, Francisco Massa, Alaaeldin El-Nouby, et al. Dinov2: Learning robust visual features without supervision. *arXiv preprint arXiv:2304.07193*, 2023.

William Peebles and Saining Xie. Scalable diffusion models with transformers. In *Proceedings of the IEEE/CVF International Conference on Computer Vision*, pp. 4195–4205, 2023.

Charles R Qi, Hao Su, Kaichun Mo, and Leonidas J Guibas. Pointnet: Deep learning on point sets for 3d classification and segmentation. In *Proceedings of the IEEE conference on computer vision and pattern recognition*, pp. 652–660, 2017.

Yichun Shi, Peng Wang, Jianglong Ye, Mai Long, Kejie Li, and Xiao Yang. Mvdream: Multi-view diffusion for 3d generation. *arXiv preprint arXiv:2308.16512*, 2023.

Yael Vinker, Ehsan Pajouheshgar, Jessica Y Bo, Roman Christian Bachmann, Amit Haim Bermano, Daniel Cohen-Or, Amir Zamir, and Ariel Shamir. Clipasso: Semantically-aware object sketching. *ACM Transactions on Graphics (TOG)*, 41(4):1–11, 2022.

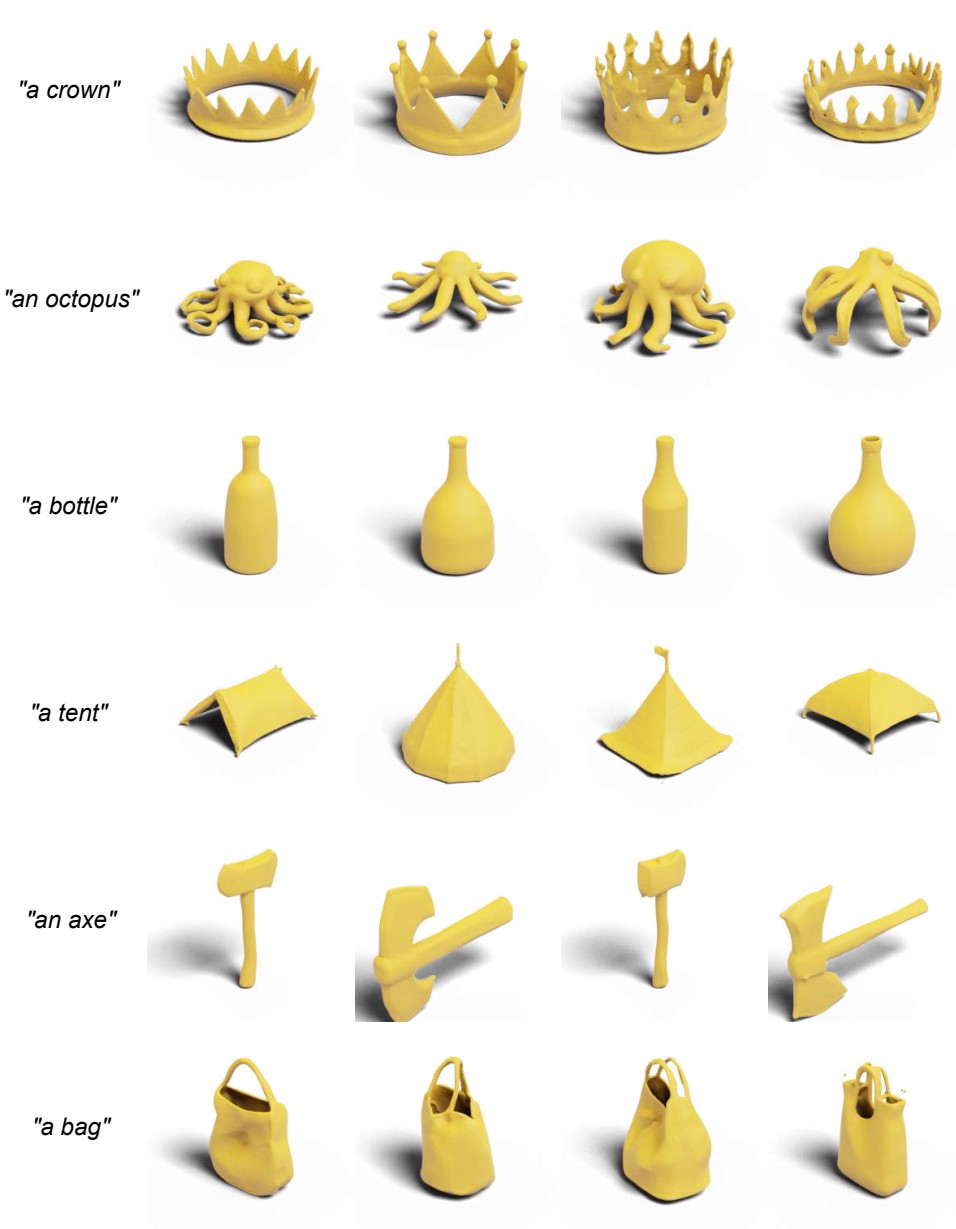

*"a crown"*

*"an octopus"*

*"a bottle"*

*"a tent"*

*"an axe"*

*"a bag"*

Figure 4: Here, for each caption, four different 3D variations are displayed. This figure emphasizes the model's flexibility in generating multiple distinct outputs for the same text description while maintaining thematic consistency.

Saining Xie and Zhuowen Tu. Holistically-nested edge detection. In *Proceedings of the IEEE international conference on computer vision*, pp. 1395–1403, 2015.