# OpenReview forum: "Wavelet Latent Diffusion (WaLa): Billion-Parameter 3D Generative Model with Compact Wavelet Encodings"
_ICLR.cc/2025/Conference — ICLR 2025 Conference Withdrawn Submission_

### Official Review · Reviewer_M8Ju · 2024-10-29

**Soundness:** 4
**Presentation:** 3
**Contribution:** 2
**Rating:** 3
**Confidence:** 5

**Summary:**

This paper introduces an extension to Make-a-Shape [1] to reduce computational cost and improve overall generative performance by incorporating a vector-quantized variational autoencoder (VQ-VAE). The pipeline consists of first converting a truncated signed distance function to a wavelet-tree representation, autoencoding this representation with a VQ-VAE that emits a low-dimensional latent grid in the bottleneck and training a latent diffusion model (LDM) in this latent space. As shown in the original LDM paper [2], this approach enables larger diffusion models, higher quality and faster inference. The authors evaluate their model on a variety of conditional generative tasks such as point cloud to mesh, single and multi-view RGB images to mesh, sketches to mesh or text to mesh. Furthermore, experiments show superior results to various previous methods, both qualitatively and quantitatively.

[1] Hui, Ka-Hei, et al. "Make-a-shape: a ten-million-scale 3d shape model." Forty-first International Conference on Machine Learning. 2024.
[2] Rombach, Robin, et al. "High-resolution image synthesis with latent diffusion models." Proceedings of the IEEE/CVF conference on computer vision and pattern recognition. 2022.

**Strengths:**

In general, the paper is well written, with a clear structure and concise visualizations. The main originality and contribution comes from the combination of the existing wavelet-tree representation with a VQ-VAE. This is a very logical step following existing approaches in diffusion-based generative literature. The VQ-VAE is trained similarly to [1] using an adaptive sampling strategy and allows the LDM to focus solely on the general shape quality and thus improving overall performance. The experimental section clearly confirms the overall performance boost in various tasks.

**Weaknesses:**

I have several concerns that are relevant for my overall rating, but the lack of contribution is certainly the most important one and is the main reason for my overall negative rating. The entire technical aspects of wavelet-based diffusion models have been explored in previous work [1, 3]. Scaling and improving diffusion models and lowering computational cost by combining VAEs and latent diffusion models has been studied extensively [2, 4, 5, 6, 7, 8, 9]. Could you clarify what the novelty is, besides improved performance and how it differs from existing approaches, besides using a VAE? While I really acknowledge and appreciate the authors' effort in implementing and training the entire model, it reads more as an extension of [1] and seems too little of a contribution for ICLR, but rather a very good fit for a journal extension. This should really not be understood as a critique of the general paper's quality, but rather about its suitability to this specific conference. Nevertheless, I am happy to be convinced of the contrary and ready to raise my rating if the authors and other reviewers provide legit reasons and arguments.
Additionally, while the experimental section is quite elaborated, it is missing basic comparisons: (1) how much improves the latent diffusion model over simply using the VQ-VAE in the unconditional setting; (2) only conditional tasks are evaluated quantitatively, but unconditional comparisons are completely missing, in particular metrics like 1-nearest neighbor accuracy (1-NNA), maximum mean distance (MMD), coverage (COV) and FID from randomly rendered images. Adding these unconditional experiments in a revised version would strengthen the evaluation and allow for comparison with existing unconditional 3D generative models.

Lastly, there are some minor weaknesses that should be easy to resolve. They can be viewed as (subjective) recommendations to (possibly) improve paper quality.
(1) The evaluation datasets are not well documented. It is not clear how many samples there are in total for each of them and how they relate to the extensive training set, e.g. how diverse are they? Could you provide the exact numbers and describe how these datasets (e.g. the categories) relate to the training set.
(2) The title and abstract suggest that the model uses a billion parameters, one page 6, line 310 it is billions of parameters but I cannot find an actual number of parameters and the actual model architecture neither in the main paper nor in the supplementary material.
(3) The section of the VQ-VAE seems to be too extensive, while the section on the wavelet-tree is definitely underrepresented and should be explained in more detail.

[3] Hui, Ka-Hei, et al. "Neural wavelet-domain diffusion for 3d shape generation." SIGGRAPH Asia 2022 Conference Papers. 2022.
[4] Vahdat, Arash, Karsten Kreis, and Jan Kautz. "Score-based generative modeling in latent space." Advances in neural information processing systems 34 (2021): 11287-11302.
[5] Sinha, Abhishek, et al. "D2c: Diffusion-decoding models for few-shot conditional generation." Advances in Neural Information Processing Systems 34 (2021): 12533-12548.
[6] Vahdat, Arash, et al. "Lion: Latent point diffusion models for 3d shape generation." Advances in Neural Information Processing Systems 35 (2022): 10021-10039.
[7] Gao, Ruiqi, et al. "Cat3d: Create anything in 3d with multi-view diffusion models." arXiv preprint arXiv:2405.10314 (2024).
[8] Zhang, Bowen, et al. "Compress3D: a compressed latent space for 3D generation from a single image." European Conference on Computer Vision. Springer, Cham, 2025.
[9] Gupta, Anchit, et al. "3dgen: Triplane latent diffusion for textured mesh generation." arXiv preprint arXiv:2303.05371 (2023).

**Questions:**

I have described the main questions, suggestions and concerns in the weakness section. I am certainly open to raising my rating if, especially, the main concern is addressed and the authors (and possibly other reviewers) can convince me of the contribution to the broad ICLR community.

Additionally, I have a few short questions:
(1) Why is WaLa depth 4 and RGB 4 of different speed in Table 3? If I understood it correctly, those models should be the same architecture wise.
(2) What is the reason for depth metrics being higher than the RGB ones?
(3) How is the ground truth data generated, i.e. the actual TSDFs from the meshes and how compute intense is this?
(4) How does your VAE fine-tuning strategy using 10k samples of each dataset actually help? If you randomly sample from each dataset, then the overall data distribution will be the same, right?

There are spaces missing on page 3 line 130, page 4 line 164, page 4 line 187 and a capital letter is used on page 8 line 425.

___

***After rebuttal***

I will keep my initial score since authors did not reply.

---

### Official Review · Reviewer_JQNj · 2024-11-03

**Soundness:** 3
**Presentation:** 3
**Contribution:** 1
**Rating:** 3
**Confidence:** 5

**Summary:**

The paper introduces Wavelet Latent Diffusion (WaLa), a method for 3D shape generation that employs wavelet-based latent encodings. By compressing a 256³ signed distance field into a 12³ × 4 latent grid, the authors claim a 2,427× compression ratio with minimal detail loss. This compression purportedly facilitates the training of large-scale generative networks, enabling the generation of high-quality 3D shapes at 256³ resolution. The models, both conditional and unconditional, contain approximately one billion parameters and are reported to generate shapes within 2–4 seconds. The authors intend to open-source the code and model weights upon acceptance.

**Strengths:**

The authors have addressed computational challenges in 3D shape generation by employing wavelet-based latent encodings. This method compresses the model size, contributing to the field by facilitating the training of large-scale generative networks. The focus on 3D generation is relevant, given the demand for high-quality 3D models in various applications. The experimental setup appears reasonable, with models containing approximately one billion parameters and generating shapes within 2–4 seconds. This indicates a practical approach to balancing model complexity and inference time.

**Weaknesses:**

•	Insufficient Justification for Wavelet Choice: The paper does not provide a compelling rationale for selecting wavelet-based encodings over other decomposition methods, such as spherical harmonics, which are commonly used in 3D shape representation.
	•	Lack of Parameter Sensitivity Analysis: There is no discussion on how sensitive the model’s performance is to the parameters controlling the wavelet transform’s frequency domain decomposition. This omission raises concerns about the robustness and generalizability of the approach.
	•	Reconstruction Accuracy Concerns: The paper does not thoroughly address potential challenges in achieving accurate reconstruction from the compressed wavelet coefficients. Without empirical evaluations demonstrating reconstruction fidelity, the effectiveness of the compression technique remains uncertain.

**Questions:**

1.	Choice of Wavelets: What is the specific rationale behind selecting wavelet-based encodings over other decomposition methods, such as spherical harmonics, for 3D shape generation?
	2.	Parameter Sensitivity: How does the model’s performance vary with different parameters controlling the wavelet transform’s frequency domain decomposition? Have experiments been conducted to assess this sensitivity?
	3.	Reconstruction Fidelity: What measures have been implemented to ensure accurate reconstruction from the compressed wavelet coefficients? Are there scenarios where reconstruction errors become significant, and how are they mitigated?

---

### Official Review · Reviewer_xqvp · 2024-11-05

**Soundness:** 3
**Presentation:** 3
**Contribution:** 1
**Rating:** 3
**Confidence:** 5

**Summary:**

The paper proposes a wavelet latent diffusion (WaLa) model for conditional and unconditional 3D shape generation. In general, the proposed method first uses wavelet tree conversion to convert a given shape, which could be of high resolution, into a wavelet tree; then, a wavelet vq-vae is utilized to reconstruct the wavelet tree. This is the first stage during which we can get latent vectors $Z$. In the second stage, a (conditional) diffusion model is adopted to generate the latent vectors $Z$ using a diffusion model. The inference is straightforward, as we first generate a latent vector using the reverse process of a diffusion model and then decode the latent vector to a 3D shape using the trained vq-vae in the first training stage. The experiments are conducted of different conditional (e.g., point cloud, images, sketches, texts, etc) and unconditional 3D shape generation tasks, and achieve the superior performance over the compared methods.

**Strengths:**

There are two key strengths of this paper:
1. As claimed by the authors, the proposed method is efficient in saving billion-parameters in 3D shape generation.
2. The quantitative and qualitative generation results are good.

**Weaknesses:**

The main concern of this paper is the novelty is very limited.
1. The wavelet tree conversion and the wavelet tree structure are proposed by [1], which is also mentioned in the paper. It is unclear if there are any improvements or modifications made to the wavelet tree proposed in [1] to adapt to the task of this paper.

2. Though parameter-saving is an advantage claimed by the authors, it is well-known that wavelet transformation can be used in signal compression by removing (certain) high-frequency coefficients. So, it weakens the contributions of this paper.

3. The vq-vae is also a well-known variant of the conventional vae, as proposed in [2]. It is also unclear whether any adaption of vq-vae has been proposed to make it more suitable for this 3D generation task.

4. The training stage 2 is nothing more than a well-known DDPM-based generation process. The only difference is not directly generating 3D shapes, but to generate wavelet coefficients. It also shares some similarities (though not exactly the same) as stable diffusion [3].

In short, this paper combines different well-known techniques without any adaptation or modification, so I failed to see any novel things in this paper, though the experimental results are good.

[1] Hui KH, Sanghi A, Rampini A, Malekshan KR, Liu Z, Shayani H, Fu CW. Make-a-shape: a ten-million-scale 3d shape model. InForty-first International Conference on Machine Learning 2024 Jan 20.

[2] Van Den Oord A, Vinyals O. Neural discrete representation learning. Advances in neural information processing systems. 2017;30.

[3] Rombach R, Blattmann A, Lorenz D, Esser P, Ommer B. High-resolution image synthesis with latent diffusion models. InProceedings of the IEEE/CVF conference on computer vision and pattern recognition 2022 (pp. 10684-10695).

**Questions:**

The method proposed in this paper is relatively straightforward, as a combination of different existing approaches, so I have no questions.

---

### Official Review · Reviewer_SWVj · 2024-11-09

**Soundness:** 4
**Presentation:** 3
**Contribution:** 4
**Rating:** 6
**Confidence:** 4

**Summary:**

This paper presents an approach to encode 3D shapes (TSDF) into a latent space using Wavelet representation via VQ-VAE. Additionally, it trains several latent diffusion models capable of generating latent codes under different conditions, such as single image, multiple images, point cloud, and voxels. The method is trained on a large dataset comprising 10 million 3D shapes collected from 19 different sources. While there are minor weaknesses in the presentation of the paper, the proposed latent wavelet tree representation is efficient and well-suited for 3D shape generation. Therefore, I recommend acceptance of the paper but reserve the right to lower the score if the concerns are not adequately addressed.

**Strengths:**

This paper proposes encoding the wavelet representation into a latent space. While the concepts of the wavelet tree and VQ-VAE latent space are not novel contributions of the paper, the efficiency of the combined solution is compelling. The authors validate the proposed representation by training a generative model, which appears to achieve good results and demonstrates training efficiency (notably, only 1 H100 GPU for training on 10 million shapes—authors, please confirm and include the training time). These aspects represent the strengths of the paper.

**Weaknesses:**

After carefully reading the submission and supplementary material, i found following weaknesses:

1. The limitations and broader impact of the proposed method are not discussed in the paper.
2. The lack of hyperlinks for citations makes reading and navigation difficult.
3. Typo: Line 130: "WaLaallows" should be corrected to "Wala allows."
4. Overclaiming in the text-to-3D capability: The method performs text-to-multiview depth generation, followed by multiview depth to 3D reconstruction. It is not a true text-to-3D model.
5. Training details are missing, including the resources and time required for the VQ-VAE model and the training time for the latent generative model.
6. The method cannot generate realistic textures, whereas the baselines in Table 3 are capable of novel view synthesis.

**Questions:**

I have several questions regarding the submission:

1. The model appears to support multiple conditioning inputs. Why doesn’t it support direct text-to-latent generation? Is this not feasible, or was it omitted due to time constraints?
2. The inclusion of SMPL and SMIL in the training dataset is confusing to me. To my knowledge, these are parametric mesh models, and the training scan data for them is not publicly available. How do they contribute to the training data?
3. The results on the anonymous page show a last update on October 10th (according to the Git commit history), which is after the ICLR deadline. Do the authors have evidence that the results were uploaded before the deadline? If the results were updated after the deadline, this would be unfair to other submissions.

---

### Note · Authors · 2024-12-06

**Comment:**

We sincerely thank all the reviewers for a thorough and constructive feedback.

While we are working on improving the current version and are considering a more relevant venue for our work, we wanted to highlight  and summarize some of the main contributions and benefits of our proposed approach, **WaLa**, as follows:
- **Relation to previous wavelet-based methods:** Building on prior wavelet-based decompositions [1, 2], we introduce an additional compression on the wavelet coefficients using VQ-VAE, which allows compressing a $256^{3}$ signed distance field into a $12^{3} \times 4 $ latent grid, achieving $2,427 \times$  compression ratio with minimal loss of detail. This efficient compression allows training large-scale generative models without increasing their inference time.
- **Large-scale 3D generative models:** By leveraging VQ-VAE compression we scale our diffusion model to 1 billion parameters while maintaining training efficiency.
- **Maintaining competitive inference times:** Despite larger models, the inference times stay highly competitive, within 2-4 seconds.
- **Better generation quality, diversity and computational efficiency:** Our method shows significant improvements across the quality and diversity of the generated samples, while maintaining computational efficiency.
- **Multiple-modalities:** We expand beyond unconditional generation and add multiple modalities such as single/multi-view images, voxels, point clouds, depth data, sketches, and textual descriptions, to showcase the versatility of our work.

Additionally, some of the other contributions include:
- Demonstrating successful scaling strategies for training on massive 3D datasets.
- Creating open-source models that can serve as strong baselines for future research.
- Contributing detailed documentation and implementation insights for the broader research community.

Through this work, we show how existing methods and approaches can be enhanced and scaled up in an efficient manner by improving the compression of the representations and by combining diffusion-based diffusion methods on these compressed representations. We hope the 3D generation community will find our work helpful and will benefit from our open-sourced large-scale models and code, which remain publicly available.

Finally, we thank the reviewers again for taking time to provide thoughtful feedback, which will help us further improve our work. \

[1] Hui, Ka-Hei, et al. "Neural wavelet-domain diffusion for 3d shape generation." \
[2] Hui KH, Sanghi A, Rampini A, Malekshan KR, Liu Z, Shayani H, Fu CW. “Make-a-shape: a ten-million-scale 3d shape model.”

**Withdrawal Confirmation:**

I have read and agree with the venue's withdrawal policy on behalf of myself and my co-authors.